# Efficiency of Printed Patient Information Leaflets Written for Total Knee and Hip Arthroplasty Patients to Reduce Their Fear of Surgery

**DOI:** 10.3390/geriatrics8050089

**Published:** 2023-09-05

**Authors:** Tünde Szilágyiné Lakatos, Balázs Lukács, Attila Csaba Nagy, Zoltán Jenei, Ilona Veres-Balajti

**Affiliations:** 1Department of Medical Rehabilitation and Physical Medicine, Faculty of Medicine, University of Debrecen, 4031 Debrecen, Hungary; jenei.zoltan@med.unideb.hu; 2Department of Physiotherapy, Faculty of Health Sciences, Institute of Health Sciences, University of Debrecen, 4028 Debrecen, Hungary; lukacs.balazs@etk.unideb.hu (B.L.); balajti.ilona@etk.unideb.hu (I.V.-B.); 3Department of Health Informatics, Faculty of Health Sciences, Institute of Health Sciences, University of Debrecen, 4028 Debrecen, Hungary; attilanagy@med.unideb.hu

**Keywords:** fear, patient education, joint endoprosthesis, rehabilitation, questionnaires

## Abstract

*Background*: Patient education plays a key role in health care. In our study, we created a new information guide for patients waiting for total knee (TKA) or hip (THA) arthroplasty. The goal of our study was to create patient education material that would reduce patients’ fear of surgery and improve their postoperative lifestyle. *Methods*: Patients in the intervention group (*n* = 44) received newly developed paper-based patient education material before surgery. The surgical fear questionnaire (SFQ) was used to assess fear reduction. A self-designed assessment questionnaire was used to measure the effectiveness of the leaflet among the intervention group patients. *Results*: The SFQ scores decreased significantly both in patients with TKA (median 37.50 IQR 30.00–40.00 vs. median 20.00 IQR 16.00–24.00) and THA (median 34.50 IQR 28.00–42.00 vs. median 20.00 IQR 16.00–22.00). A control group with TKA (median 37.50 IQR 30.00–40.00 vs. median 64.50 IQR 54.00–82.00) and THA (median 34.50 IQR 28.00–42.00 vs. median 73.00 IQR 56.00–81.00) was also included. An assessment of the content, usability, and clarity of the new leaflet showed that patients rated the new leaflet as almost entirely usable (median score 12.00–10.00). *Conclusions*: Our results suggest that new printed patient education material may reduce the fear of surgery.

## 1. Introduction

In high-quality healthcare systems, patients must be provided with accurate and comprehensive information. In 2015, the World Health Organisation developed a global strategy that highlights the importance of people-centred and integrated care. Currently, healthcare policy aims to increase health gains cost-effectively [1,2]. In line with this approach, patient education has become a central issue. High-quality patient education is indispensable in all areas of healthcare. According to the literature, the preparation of patients for procedures has a positive effect on the outcome of scheduled treatments and operations [3,4]. Patients who understand and are familiar with the process of looking after themselves can become more actively involved in their treatments. This may lead to a more successful outcome and higher health gains for both the patient and the healthcare system. According to Marc O’Reilly et al., patient education before hip and knee arthroplasty results in shorter time spent in the hospital and an improvement in the outcome of rehabilitation, leading to an improvement in cost-effectiveness [3]. Several previous studies have shown an upward trend in TKA and THA surgery in both OECD and Eastern European countries. The prevalence of both surgical interventions is significantly higher in women. These data also underline the growing importance of high-quality patient education in the future, which could reduce the number of postoperative days spent in the hospital and thus reduce the workload of the healthcare system [5,6,7].

Patient education material has various forms, including DVDs, information guides, and brochures. Patients like preoperative group education and also like to talk to other patients who have already undergone what they are facing now. They are more willing to accept information from other patients because they find it far more realistic and genuine. Many patients search the internet for information; men even like to watch surgeries, while women definitely refuse to do so [8,9]. According to Tessa Dekkers et al., online patient education can be time- and cost-effective if the objective is to improve patients’ levels of knowledge and satisfaction. However, these results are not necessarily representative of the entire orthopaedic patient population and apply only to younger, highly educated patients who are regular internet users [10]. The main goals of our study were to reduce patients’ fear of surgery and postoperative rehabilitation using a patient information guide that was adapted to the comprehension skills of patients who were supposed to undergo total knee arthroplasty (TKA) and total hip arthroplasty (THA) surgery and to assess the usability and efficiency of our patient information guide. Our premise was that our patient guide would provide patients with new and sufficient information, contribute to reducing their fear of surgery, and help them prepare themselves for their new postoperative lifestyle.

## 2. Methodology

In the framework of our research, a new patient information guide was introduced at the Kenézy Gyula Hospital of the University of Debrecen (DE KEK). Patients were provided with detailed written information on the procedure of total knee and hip arthroplasty, on the importance of postoperative rehabilitation, and on movements that they should avoid after surgery.

Target Group

Our target group was comprised of patients scheduled for THA and TKA at the Department of Traumatology and Hand Surgery of the Kenézy Gyula Hospital of the University of Debrecen.

Inclusion criteria were as follows:-The study included patients who had been booked for total knee arthroplasty and total hip arthroplasty surgery, and they had to agree to review our educational material and complete questionnaires about the evaluation of the educational booklet and the SFQ questionnaire, which assesses fear of the surgical process.

Exclusion criteria were as follows:-Patients who chose not to complete questionnaires.

### 2.1. Study Process

The study period was from 1 February 2023 to 31 May 2023. We started our quasi-experimental study with a consultation two months before surgery, when the specialist agreed on the surgery date with the patient. At this time, all patients included in the study were asked to complete a surgical fear questionnaire (SFQ) [11]. Subsequently, half of the patients received a patient information leaflet (the intervention group), while the other half did not (the control group). The groups’ (intervention and control) age- and gender-matched controls were randomly selected from cases (whose sample size was estimated a priori). The educational material was specifically designed for patients undergoing hip and knee replacements. Patients were then prepared for surgery and the postoperative period in their homes, with (the intervention group) or without (the control group) studying the patient information leaflet.

On arrival at surgery, before the procedure, both the intervention group and the control group completed the SFQ questionnaire again to assess the extent to which the information leaflet had influenced the patients’ fear of surgery. After surgery, patients were admitted to the rehabilitation department of the university. At the end of the clinical rehabilitation period, patients in the intervention group were asked to complete a self-report questionnaire about the usefulness of the patient information leaflet and the content and comprehensibility of the new information given to them (Figure 1).

The estimated sample was calculated using the “sampsi” command of Intercooled Stata v13. The type I error was set to 5%, and the statistical power was set to 90%. The effect size was evaluated by Cohen’s d. The refusal rate was also taken into consideration. The calculation was based on relevant articles [8,9].

### 2.2. Patient Information Guide

TKA and THA patients go through a long and painful period before their joint replacement. They have difficulty walking, climbing stairs, and other limitations in everyday life. Sometimes, a well-designed kinesiotherapy can postpone surgery for some time, but in most cases, arthroplasty remains the only option to ease the pain. The surgery relieves patients’ pain and ensures improvements in their functions. We created separate printed patient information guides for TKA and THA patients. The purpose of the guide was to educate patients on the procedure, inform them of what they have to do before and after surgery, and help our patients build an appropriate lifestyle after surgery. The patient education material did not include a diagram; we preferred the text-based guide.

The guide covered four main areas:The first part described knee and hip prosthesis types, the expected lifetime of replacements, and the factors that affect their lifetime.The second part focused on the importance of preoperative programmes. In order to avoid pain, patients load their healthy side more, resulting in a changed posture. Patients do not use the appropriate muscles while walking, which results in a decrease in muscle strength and muscle tone on the painful side, which in turn makes postoperative rehabilitation longer and more difficult. In this part, we also informed patients that preoperative exercises under the supervision of a physiotherapist could help strengthen or, if necessary, relax the affected muscles around the joint to be replaced, which may shorten the rehabilitation period after surgery. They can also learn exercises that we deem important to start immediately after the surgery. Patients were also informed about the walking aids that they would have to use after surgery if they decided to participate in our program. The patient education material also included information on where our preoperative programme would be available for patients waiting for hip and knee arthroplasty surgeries.In the third part, our patients received useful advice on lifestyle, such as how to get out of bed after surgery and how to use a high bed and chair after surgery. The information also highlighted the use of a burlap or elastic bandage. The leaflet also described the near and far aids to use after surgery. It also explained how to transport patients to their homes immediately after surgery or after postoperative rehabilitation. They also described options for patients to continue regular exercise and sports activities (swimming, cycling, or driving) after the postoperative rehabilitation period.

The information leaflet for hip replacements also recommends putting a firm, long, large pillow between the knees after the operation.

Patients waiting for a hip replacement should be made aware that before the operation they should find out about the movements that should not be used at all after the operation (forbidden movements):Intensive stretching of the hip, leaning forward (if it is necessary to bend down, slide your operated leg backwards extended, and bend your healthy leg);Crossing legs in any posture;Supination and pronation of the operated leg with straight legs;Twisting the torso while standing with straight legs (the leg must be moved in the direction of the rotation in sync with the body).



4.The fourth part offered a general description of postoperative rehabilitation, which is further divided into two stages: early and late rehabilitation. The former begins on the day of the operation at the Department of Traumatology. Based on the recommendations of the surgeon in terms of load, a physiotherapist teaches patients how to do tailored exercises and how to use their mobility aids. In the late rehabilitation stage, when a patient has left the department where the operation was performed, rehabilitation may be continued either at the Department of Rehabilitation or at the patient’s home in the form of home patient care, based on the recommendation of the specialist who performed the surgery. The first phase of late rehabilitation can be carried out more efficiently in a hospital setting because of the availability of equipment that supports the restoration of joint functionality and facilitates a quicker recovery.


The first three pages of the four-page, A5-size guide provided patients with new information, while for those patients who did not understand something (patients with learning difficulties) about the patient education material, contact details of the physiotherapists of the department and the investigator in charge of the study were provided so that any questions about the patient information leaflet could be answered.

### 2.3. Data Collection

#### 2.3.1. Questionnaires Used for Data Collection

##### Surgical Fear Questionnaire (SFQ)

The SFQ, which has been used since 2011 and is also validated in the Hungarian language [11], contains 10 questions. Patients were asked to indicate the level of their fear on a 0–10 scale. Zero meant no fear, and 10 represented severe fear. The questionnaire yields a total score of 0 to 100. The lower the score, the less the patient is afraid of complications.

##### Self-Designed Questionnaire to Measure the Effectiveness and Usability of the Patient Information Leaflet

We used a questionnaire of our own design to assess the usability and comprehensibility of the leaflet. It included 12 closed-ended questions and one open-ended question related to the four sections of the guide. Based on the answers, we gathered information on the amount of new information provided by the leaflet, its comprehensibility, and its usability.

Patients had three options to choose from with every question and could give a maximum of 12 points, which meant that the guide was easy to understand and provided new information.

Patients completed the questionnaire on a voluntary basis at the end of the postoperative rehabilitation programme.

We helped patients understand the questions when necessary.

### 2.4. Statistical Analysis

The statistical analysis was performed using Microsoft Office Excel and SPSS software. Statistical analysis to evaluate differences was performed using an unpaired (independent) *t*-test or two-sample Wilcoxon rank-sum (Mann–Whitney) test. The normality test was performed using the Shapiro–Wilk test. The results were considered significant if *p* < 0.05.

The results were broken down by prosthesis type (total hip and total knee) and age group (under 70 and over 70). We chose 70 years of age because, even though retirement age is 65 in Hungary, due to the social and economic situation, many people keep working as pensioners, full-time or part-time, until the age of 70 [12,13]. This raises the important question of whether still-active patients will be able to continue working and, if so, how soon after the surgery.

### 2.5. Ethical Approval

The study was approved by the Research Ethics Committee of the Kenézy Gyula Hospital of the University of Debrecen (É/23 January 2020), and participants gave informed consent to the collection and processing of data.

## 3. Results

### 3.1. Demographic Data

A total of 88 people participated in the study. We examined 44 patients in both control (27 females and 17 males, 15 TKA and 29 THA) and intervention (30 females and 14 males, 14 TKA and 30 THA) groups.

### 3.2. Surgical Fear Questionnaire (SFQ)

This questionnaire was used to investigate the effect of the patient information leaflet on reducing preoperative fear. It is innovative if completed by the patient in the preoperative period. Therefore, patients completed the questionnaire before surgery, during the initial consultation, and at the time of admission. The questionnaire was completed by 88 patients (intervention and control groups).

In the intervention group, the SFQ scores of patients who underwent total knee replacement (TKA) were significantly lower (*p* < 0.001, Cohen’s d = 1.76) after reading the patient information leaflet (median 20.00 IQR 16.00–24.00) compared to the baseline patient population (median 37.50 IQR 30.00–40.00). Patients who underwent total hip replacement (THA) also had significantly lower SFQ scores (*p* < 0.001, Cohen’s d = 1.76) after reading the patient information leaflet (median 20.00 IQR 16.00–22.00) compared to all patients (median 34.50 IQR 28.00–42.00) at the first SFQ assessment. (Table 1).

In the control group, the SFQ scores of patients who underwent total knee replacement (TKA) but did not receive the information leaflet were higher (median 64.00 IQR 54.00–82.00) compared to the baseline patient population (median 37.50 IQR 30.00–40.00), (*p* = 0.005). In the control group, the SFQ scores of patients who underwent total hip replacement (THA) but did not receive the information leaflet were higher (median 73.00 IQR 56.00–81.00) compared to the baseline patient population (median 34.50 IQR 28.00–42.00), (*p* < 0.001) (Table 1).

In the intervention group (TKA and THA, *n* = 44), the SFQ score was significantly (*p* < 0.001, Cohen’s d = 1.76) lower (median 20.00 IQR 16.00–24.00, median 20.00 IQR 16.00–22.00) compared to the control group’s (TKA and THA, *n* = 44) score (median 64.50 IQR 54.00–82.00, median 73.00 IQR 56.00–81.00) (Table 1).

We also examined the fear of surgery among patients under and over 70 years of age. We found that the level of fear significantly decreased in both age groups. Table 1 shows data related to fear of surgery and age.

After studying the effect of the information leaflet on the TKA and THA patient groups, the possible influencing factors of the SFQ were also evaluated. As you can see in Table 2, patients who are of an older age, female, and without the information leaflet were found to have a higher SFQ score (Table 2).

### 3.3. Evaluation of the Patient Information Leaflet in Terms of Clarity, New Information Content, and Usefulness

A total of 44 patients completed the questionnaire. The proportion of women among the TKA patients (*n* = 14) was 78.6%, and their mean age was 71.5 ± 7.33. The proportion of women among the THA patients (n = 30) was 63%, and their mean age was 66.23 ± 12.50.

Evaluation questions

Evaluation of the new patient information guide showed that the median score of the 44 respondents was 12 on all three questions relating to how easy it was to comprehend the content, how much new information the guide contained, and how useful it was in the postoperative phase.

Results broken down by prosthetic types and age:

In terms of comprehensibility, TKA patients (*n* = 14, 78.6% females) achieved a median score of 12 points. No statistical difference was evaluated between age groups.

Regarding new information content, the group achieved a median score of 12.00.

Regarding the usefulness of the guide, the median score was 12. The difference between the scores of patients under and over 70 years of age was not significant.

The median score for the comprehensibility of the guide in the group of THA patients (*n* = 30, 63% females) was 12.

Regarding new information content, the median score in the THA group was 11.50. Patients under the age of 70 scored higher (median 12.00, IQR 11.00–12.00 vs. median 10.00, IQR 9.75–12.00), but no statistical difference was evaluated between age groups.

With respect to the usability of the guide, the median score was 12 in this group. The median score in the under-70 age group was 12, compared to 11.00 in the over-70 age group; the difference was not statistically significant. Table 3 shows the data relating to the efficiency of the guide.

## 4. Discussion

The aim of our study was to mitigate fear of the surgery in THA and TKA patients using our self-designed patient education material and to assess the usefulness of our written patient information guide designed for the preoperative and postoperative periods. The results demonstrated that, due to its clarity, content, and usability, the guide developed in our study significantly reduced the patients’ fear of surgery. Patients involved in the study understood that they could do a great deal to recover quicker and prevent complications, which often leads to lower healthcare costs, as suggested by the literature data [3].

When we designed the programme, we made the assumption that the information that we produced would reduce patients’ fear of surgery, and we were successful in confirming this. We obtained significantly lower scores for both types of prostheses based on the surgical fear questionnaire (SFQ). Patient information and face-to-face interviews reduced the patients’ fear of surgery. The results of the control group show that as the date of surgery approaches, the patient becomes more afraid of the procedure. They are less anxious two months before surgery than when they go to the operating room the day before surgery. After the consultation with the doctor, patients were more relaxed, according to the SFQ survey, because they were able to discuss everything with the doctor about the time ahead. In the two months between the consultation and the surgery, they did not have the opportunity for another consultation and were not given written patient education material to read more than once, so it is understandable that they were more anxious immediately before the SFQ survey. All participants found our patient information guide easy to understand. The median score was 12 in both groups, which means that there were only a few respondents who did not understand some part(s) of the content. Our guide designed for patients scheduled for hip and knee arthroplasty provided new information for both groups of patients.

Previous research projects have established that patient information guides should be designed for a sixth-grade reading level in terms of comprehensibility [14,15,16]. Based on the results, our self-designed guide complied with this requirement, as the patients involved understood the content and were able to use the information. In line with the findings of Kennedy et al., participating patients liked written information because they did not have to memorise it and could read it again at any time [2]. In general, we can claim that appropriate patient information guides play an important role in patient education. This is especially true in light of the fact that society is ageing faster, which means that an increasing number of older people will need THA or TKA [17]. Rohringer et al. found that the level of patients’ health literacy plays a key role in patient education in the orthopaedic rehabilitation period following THA or TKA [18]. Bitzer et al. reported that, as a result of patient education in hospital settings, health literacy levels could be effectively increased, enabling patients to contribute to their recovery in their own rehabilitation process [19]. Rackwitz et al. found that comprehensive education before the intervention plays a central role in the preparation of THA and TKA patients and also facilitates recovery after surgery. They concluded that preoperative education may reduce fear of intervention and minimise the need for painkillers after surgery [17]. Clarius et al. came to similar conclusions. They held interdisciplinary sessions for patients and informed them about their condition, the intervention, pain management, early mobilisation, and the stages of the patient route. Families and friends were also involved in the programme so that they could support the patient before and after the surgery, as well as during the rehabilitation period [20]. In 2012, Tullamore Midlands Regional Hospital in Tullamore introduced a “common school” for patients waiting for surgery. Patients were educated in 90-min sessions. The multidisciplinary programme consisted of four integrated sessions combining PowerPoint presentations, educational videos, and model demonstrations involving nurses, anesthesiologists, surgeons, and physiotherapists. They were given a guide explaining all aspects of the intervention. Their family members were also able to attend the sessions, reducing anxiety and helping patients absorb the information. They compared data collected during the first session with data recorded after the last session (*p* < 0.001) [3]. Pierluigi Sinatti and colleagues reported on how patient education affected pain levels and the process of regaining knee and hip joint function. Patient education significantly reduced postoperative pain levels in 84% of total hip arthroplasty (THA) and total knee arthroplasty (TKA) patients that they studied [4]. One of the basic principles of patient education is that the content of patient information leaflets should always be adapted to the patients’ intellectual abilities [14,21]. Based on their review of existing leaflets, Stenquist et al. concluded that although they were only designed for the eighth-grade reading level, they were not suitable for conveying information [15]. Patient information guides designed for the fifth-grade reading level were found to be suitable for patient education [22]. Several health care facility administrators have recommended that patient information guides not exceed the sixth- to eighth-grade reading level. It should be noted, however, that the requirement that the patient information guide be easily understood by anyone also limits the delivery of important and accurate information [21]. Sentence structure is also very important in patient education materials. Majid and colleagues reviewed the importance of preoperative patient education in orthopaedics [15]. In 2013, Yi and colleagues analysed articles on patient education on the websites of five leading orthopaedic implant companies. Each article was measured using the Flesch–Kincaid (FK) readability test. They counted articles whose readability was below grade eight (the average level for US adults) and those below grade six (the recommended level for patient education). Most of the guides produced by implant companies were too complicated for the average patient. The authors suggested that more effort should be made to improve the readability of orthopaedic patient information leaflets [16]. The literature suggests that the majority of the population may have difficulty understanding patient pathways [21].

In our study, the compilation of the written patient education material was based on the results of previous research. Taking the older generation’s needs into consideration, the patient education material was at the sixth-grade reading level and was paper-based. In the process of developing this patient education material, our aim was to make it as easy to comprehend as possible while still containing all the specific information that is useful during the pre- and postoperative periods. Ineffective patient pathways also have significant economic consequences; the health care costs of the undereducated are nearly four times higher than those of the medically literate [14,21]. Patient education can also aid recovery after surgery. According to Koivisto et al., preoperative education prepares patients for postoperative care, but the different needs of patients for patient education need to be taken into account [23]. Doinn et al. conclude that health education is essential for orthopaedic patients. They found that patients who underwent surgery without health literacy did not understand the procedure and did not follow preoperative instructions, resulting in more postoperative complications. Their conclusions emphasised the importance of patient education [24]. Poor or misunderstood communication between doctors and patients can increase the risk of medical malpractice, while effective communication reduces anxiety, improves patients’ attitudes towards the procedure, and can have a positive impact on clinical outcomes [21].

As we have outlined above, appropriate patient information is an indispensable element of modern health care systems. Its advantages are evident on patient, health care system, and societal levels alike. According to the Hungarian Health Care Standard, which entered into force on 31 January 2007, health care institutions providing inpatient care must employ professionals who are authorised to inform patients and to obtain and record patients’ consent in writing [25].

## 5. Limitations

Unfortunately, the fear of surgery questionnaire was completed by very few people, based on the number of patients included.

The self-designed questionnaire was not a validated questionnaire. It was meant to examine the effect of a single intervention, which is why we did not have a validated questionnaire at our disposal for this purpose. In compiling the questionnaire, we relied on questions used previously in scientific examinations in Hungary (ELEF). We had this questionnaire reviewed by specialists in several fields (medicine, physiotherapy, and patient education) before distributing it among patients.

We did not measure access to “online” information in either group, as our patients were in the older age group and older people in our country use the internet less.

## 6. Conclusions

Our research results confirmed that the patient information guide designed for THA and TKA patients effectively reduced their fear of surgery. Our conclusion is that patient education should be integrated into the preparation of patients for procedures (surgeries and treatments) where patient preparation is possible.

We continue to regard it as our priority to distribute our guide, which has been updated based on our findings, to more and more patients. The guide increased the efficiency of the treatment of THA and TKA patients, who were provided with detailed information on each step of the procedure, which helped them recover quicker. We are planning to continue our research with additional studies.

## Figures and Tables

**Figure 1 geriatrics-08-00089-f001:**
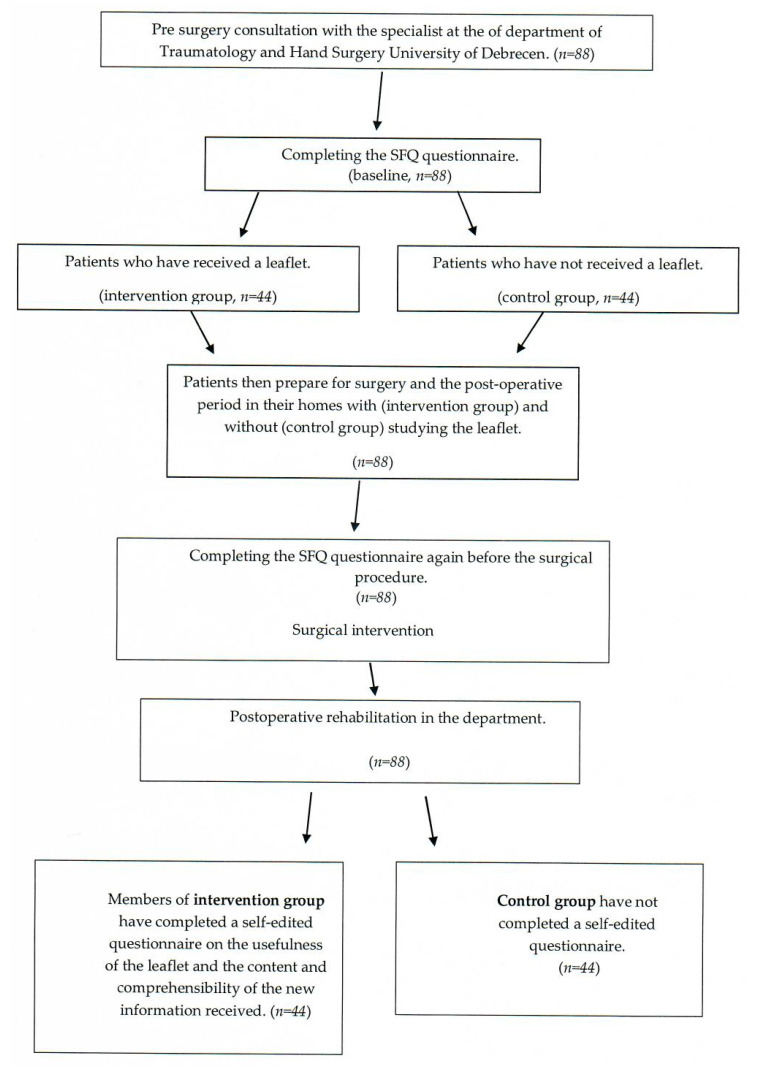
Flowchart of the study.

**Table 1 geriatrics-08-00089-t001:** Surgical fear questionnaire results: baseline, control, and intervention groups. The table shows data on fear of surgery in the baseline and intervention groups and in patients under 70 and over 70.

		Baseline			Control Group				Intervention Group			
*Surgery*	*Group*	*Median*	*IQR*	*p-value (age under 70* vs. *70 and over)*	*Median*	*IQR*	*p-value (age under 70* vs. *70 and over)*	*p-value (baseline* vs. *intervention group)*	*Median*	*IQR*	*p-value (age under 70* vs. *70 and over)*	*p-value (baseline* vs. *intervention group)*
*Total knee arthroplasty*	under 70	38.00	30.00–40.00	0.835	54.00	20.00–69.00	0.097	0.379	22.50	19.00–28.00	0.298	0.058
	70 and over	36.50	31.50–39.50		75.00	63.00–84.00		0.002	17.50	15.50–23.50		0.001
	total	37.50	30.00–40.00		64.50	54.00–82.00		0.005	20.00	16.00–24.00		<0.001
*Total hip arthroplasty*	under 70	35.00	28.00–42.00	0.992	56.00	31.00–71.00	0.008	0.016	20.00	16.00–22.00	0.719	<0.001
	70 and over	33.50	27.50–43.00		77.00	73.00–86.00		<0.001	20.00	15.50–23.50		<0.001
	total	34.50	28.00–42.00		73.00	56.00–81.00		<0.001	20.00	16.00–22.00		<0.001

**Table 2 geriatrics-08-00089-t002:** Identifying the possible influencing factors of SFQ.

	**Total Hip Arthroplasty**		**Total Knee Arthroplasty**	
	coeff [95% CI]	*p*	coeff [95% CI]	*p*
Age (years)	−0.05 [−0.34–0.23]	0.711	1.23 [0.02–2.44]	0.046
Gender (female/male)	4.71 [−1.4–10.81]	0.128	17.34 [2.19–32.48]	0.027
Group (control/intervention)	55.15 [50.14–60.16]	<0.001	49.5 [34.92–64.08]	<0.001

**Table 3 geriatrics-08-00089-t003:** Evaluation of the patient information guide. The table shows data regarding the comprehensibility, new information content, and usability of the patient information guide for patients under and over the age of 70 who underwent THA (*n* = 30) and TKA (*n* = 14).

		Comprehensibility		New Information		Usefulness	
Surgery	Group	Median	IQR	Median	IQR	Median	IQR
Total knee arthroplasty	total	12.00	12.00–12.00	12.00	11.25–12.00	12.00	12.00–12.00
	below 70	12.00	12.00–12.00	12.00	10.50–12.00	12.00	11.25–12.00
	over 70	12.00	12.00–12.00	12.00	11.00–12.00	12.00	12.00–12.00
Total hip arthroplasty	total	12.00	11.25–12.00	11.50	10.00–12.00	12.00	11.00–12.00
	below 70	12.00	12.00–12.00	12.00	11.00–12.00	12.00	11.00–12.00
	over 70	12.00	11.00–12.00	10.00	9.75–12.00	11.00	10.25–12.00

## Data Availability

Data are available from the authors at reasonable written request after authorization by the Data Protection Office of the University of Debrecen, Hungary.

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
