# Peer review of "Efficiency of Printed Patient Information Leaflets Written for Total Knee and Hip Arthroplasty Patients to Reduce Their Fear of Surgery"

_geriatrics, 2023, doi:10.3390/geriatrics8050089_

Round 1

Reviewer 1 Report

Efficiency of printed Patient Information Leaflet written for total knee and hip arthroplasty patients to reduce their fear of surgery.

The authors aimed to establish whether giving patients printed educational material prior to Total Knee Arthoplasty (TKA) and Total Hip Arthroplasty (THA) surgery, might reduce their fear of surgery; which might then in turn help with their post-operative recovery

The Surgical Fear Questionnaire (SFQ) was used to assess fear reduction in the 44 Study patients who received printed information, and 44 “Controls”, who did not.  Patients also completed a “self-designed” assessment questionnaire to measure the effectiveness of the leaflet.

The authors found that the SFQ scores decreased significantly in TKA who had been given the printed material (median 37.50 IQR 30.00-40.00 vs. median 20.00 IQR 16.00-24.00) and similarly in the THA patients (median 34.50 IQR 28.00-42.00 vs. median 20.00 IQR 16.00-22.00).

The patients rated the information booklets as “almost entirely usable”.

The authors concluded that the introduction of their new printed patient education material appeared to reduce the fear of surgery, and they considered that it was an important adjunct to the pre- and post-operative periods.

I have some comments

1)

Giving patients educational material before surgery is not a novel concept and is part of the standard informed consent process before surgery.  In the UK patients also attend a pre-operative “joint replacement school” where they all receive further pre-operative education, as part of the our routine Enhanced Recovery Pathway. This has been shown to reduce anxiety, to set realistic surgical expectations, and leads to a reduced hospital length of stay.

2)

One might therefore question whether the authors behaved ethically by NOT providing the control group with any printed information prior to the surgery.

Nevertheless this report is a useful corroboration that the provision of some pre-operative educational material reduced the level of surgical fear within their TKA/THA patient cohort.

3)

How were the patients randomised to each group?

4)

Lines 61-67 (Methodology). Can the authors state the study time period, dates etc, how long before surgery these printed material were given, and how long after surgery the patients completed their questionnaires.

5)

Did the printed information contain diagrams?

6)

Presumably the Control group (and Study group) would have received all this same information verbally in the outpatient department when they were booked for surgery? If not, then what information were these 2 groups of patients routinely given?

7)

Is it ethical to have withheld the information in the Second Part from the patients in the control group? This information would have improved their pre-operative physical health in the run up to surgery.

8)

Was there any metric for patient understanding of the information; for example testing the patients on what information they retained with an MCQ? How much had they recalled at the time of their surgical admission? How many actually read the document?  

9)

How would the authors tackle patients with learning difficulties?

10)

It would have been useful to also learn whether their intervention had any other benefits, for example earlier mobilisation, reduced analgesic requirements, reduced hospital length of stay etc….

11)

In the abstract the authors have stated that the SFQ scores decreased significantly in TKA/THA who had been given the printed material, please can they also state the scores of the Control group in the Abstract.

12)

The results indicate that the authors’ standard process of consenting may be insufficient given how anxious their Control patients appear to be. I would be concerned as a clinician that I was not explaining the process sufficiently.

13)

I am also surprised that none of the Intervention patients appeared to remain at all anxious prior to their surgery (even after receiving the printed information); while virtually All the control patients became more fearful before surgery. This seems out of keeping with my own observations where one might expect some persistent anxiety in some patients and no anxiety in others; depending on their personality type and a myriad of other factors.

14)

Tables 1 and 2 show that the Baseline scores of the 2 groups were the same. This is highly improbable. I assume that these are pooled results. Please separate the baseline scores and state them for each of the subgroups, rather than pooling the whole series.

15)

Table 3 is duplicating Tables 1 and 2 to a large degree. There might be a better way of demonstrating these statistical differences between the 2 groups.

16)

All the patients might have accessed other “online” information from other sources and toher countries. This is fairly routine behaviour and many of our own patients come to the clinic with printouts of information they have downloaded from all kinds of websites; including endorsed information booklets.. The authors do not appear to have factored this possibility into their analyses when comparing their 2 cohorts.

Some minor editing required

Author Response

Response to Reviewer 1 Comments

Dear Reviewer! Thank you for your careful review, for your important comments and suggestions and for giving us the opportunity to resubmit a revised version of the manuscript “Efficiency of printed Patient Information Leaflet written for total knee and hip arthroplasty patients to reduce their fear of surgery” for publication in the Special Issue of Geriatrics.

Point 1:

Giving patients educational material before surgery is not a novel concept and is part of the standard informed consent process before surgery.  In the UK patients also attend a pre-operative “joint replacement school” where they all receive further pre-operative education, as part of the our routine Enhanced Recovery Pathway. This has been shown to reduce anxiety, to set realistic surgical expectations, and leads to a reduced hospital length of stay.

Answers to the questions:

  • Of course, in Hungary too, it is mandatory to inform patients before the surgical procedures. In the case of the patients included in the study, both the intervention and the control groups received the standard informed consent and the patients signed a form to confirm that they had been informed, but this standard guide informed the patients only about the risks of surgical procedure, and did not contain any relevant informations about the post-operative lifstyle.
  • The novel of our research was a preparation of a paper based patient education guide, which contained informations about the advantages of the surgery, the process of rehabilitation, and patient education related to the post-operative lifestyle. Unfortunately, it is not common practice in Hungary to provide patient education material before surgery. At the Kenézy Gyula Hospital of the University of Debrecen, there was no such practice in the patient group under study, which is why we prepared such patient education material for the patients waiting for knee and hip replacement. Our study had the same results as you describe, a reduced fear of the procedure for patients who received patient education material.

Point 2:

One might therefore question whether the authors behaved ethically by NOT providing the control group with any printed information prior to the surgery.

Nevertheless this report is a useful corroboration that the provision of some pre-operative educational material reduced the level of surgical fear within their TKA/THA patient cohort.

Answers to the questions:

  • As described in our previous answer (Point 1), in Hungary information to patients is mandatory, but the form (oral, written) is not regulated. On the basis of international examples and scientific studies, more and more Hungarian institutions are nowadays introducing the use of written patient information material.
  • In order to prove that our patient education material was useful for the patient group, we had to set up a control group, which meant that one of our groups did not receive any educational material. Following the completion of the study, the positive results of the study have led to all patients (TKA, THA) receiving patient education material to prepare for surgery and subsequent rehabilitation.

Point 3:

How were the patients randomised to each group?

Answers to the questions:

Age and gender matched controls were randomly selected to cases (whose sample size was estimated a prior).

Point 4:

Lines 61-67 (Methodology). Can the authors state the study time period, dates etc, how long before surgery these printed material were given, and how long after surgery the patients completed their questionnaires.

    Answers to the questions:

The study period was from 01.02.2023 to 31.05.2023. Patients in the intervention group received the educational material at their medical consultation, 2 months before surgery. At this time, all patients (control, intervention) who participated in the study completed the SFQ questionnaire for the first time, and the second SFQ questionnaire was completed at the time of admission the day before surgery. The questionnaire measuring the usefulness of the patient education material was completed by the intervention group at the end of post-operative rehabilitation in hospital.

Point 5:

Did the printed information contain diagrams?

Answers to the questions:

The patient education material did not include a diagram, we preferred the text based guide.

Point 6:

Presumably the Control group (and Study group) would have received all this same information verbally in the outpatient department when they were booked for surgery? If not, then what information were these 2 groups of patients routinely given?

Answers to the questions:

As we described in our previous answer (Point 1), the surgeon who performed the surgery, verbally informed both groups (control, intervention) about the riscs of surgical procedure in the outpatient department when they were booked for surgery. Both groups received the same basic information during the verbal debriefing.

Point 7:

Is it ethical to have withheld the information in the Second Part from the patients in the control group? This information would have improved their pre-operative physical health in the run up to surgery.

Answers to the questions:

In Hungary the patient information in the hospitals before any surgery is mandatory but very poor. In most cases the physicians have to perform a standard oral information about the riscs of the surgery and the patients signed a form to confirm that they had been informed. This is the standard process, and other informations are not withhold, but these are not the part of the basic patient information.

We agree with your point, that more information would have improved the patients’ pre-operative physical health in the run up to surgery, that is the reason why we made the patient information leaflet and tried to build it into the pre-operative process.

Point 8:

Was there any metric for patient understanding of the information; for example testing the patients on what information they retained with an MCQ? How much had they recalled at the time of their surgical admission? How many actually read the document?  

Answers to the questions:

We could only assess text comprehension using the questionnaire method presented in our study. In the questionnaire, we asked the patients the questions described in the study (about the usefulness of the patient information leaflet and the content and comprehensibility of the new information given to them). All the participants of intervention group read the education material according to their own admission.

.Point 9:

How would the authors tackle patients with learning difficulties?

Answers to the questions:

In preparing the patient education material, we have taken into account a number of research studies, which we have included in this article. In Hungary, the school age is 16 years old, so the wording of the educational material was set at the level of a pseudo-lower secondary school at grade 6, so that everyone can understand the content. For those patients who did not understand something about the patient education material, contact details were provided: the physiotherapists of the department, the investigator in charge of the study, so that any questions about the patient information leaflet could be answered for patients.

Point 10:

It would have been useful to also learn whether their intervention had any other benefits, for example earlier mobilisation, reduced analgesic requirements, reduced hospital length of stay etc….

Answers to the questions:

In a previous study we showed that, as a result of the prehabilitation program, the length of postoperative hospital stay decreased (THA: median 31.5 (IQR 26.5–32.5) vs. median 28 (IQR 21–28.5), TKA: median 36.5 (IQR 28–42) vs. median 29 (IQR 26–32.5)), and the patients’ quality of life showed a significant improvement (TKA: median 30.5 (IQR 30–35) vs. median 35 (IQR 33–35), THA: median 25 (IQR 25–30) vs. median 33 (IQR 31.5–35)). The flexion movements were significantly improved through prehabilitation in both groups.

Szilágyiné Lakatos T, Lukács B, Veres-Balajti I. Cost-Effective Healthcare in Rehabilitation: Physiotherapy for Total Endoprosthesis Surgeries from Prehabilitation to Function Restoration. Int J Environ Res Public Health. 2022 Nov 16;19(22):15067.

Point 11:

In the abstract the authors have stated that the SFQ scores decreased significantly in TKA/THA who had been given the printed material, please can they also state the scores of the Control group in the Abstract.

Answers to the questions:

We have made the necessary changes in the Abstract.

Point 12:

The results indicate that the authors’ standard process of consenting may be insufficient given how anxious their Control patients appear to be. I would be concerned as a clinician that I was not explaining the process sufficiently.

Answers to the questions:

The results of the control group show that as the date of surgery approaches, the patient becomes more afraid of the procedure. They are less anxious 2 months before the operation than when they go to the surgical ward the day before the operation. After the consultation with the doctor, patients were more relaxed according to the SFQ survey, as they were able to discuss everything with the doctor about the time ahead. In the two months between the appointment and the surgery, they did not have the opportunity for another consultation and were not given patient information leaflet to read more than once, so it is understandable that they were more anxious immediately before the SFQ survey.

Point 13:

I am also surprised that none of the Intervention patients appeared to remain at all anxious prior to their surgery (even after receiving the printed information); while virtually All the control patients became more fearful before surgery. This seems out of keeping with my own observations where one might expect some persistent anxiety in some patients and no anxiety in others; depending on their personality type and a myriad of other factors.

Answers to the questions:

On one hand the reason of the too little deviation among our results can be the consequence of the low sample size. On the other hand, several previous studies have reported, that the usefulness of a patient information leaflet significantly reduces patients' fear of surgical procedure. In addition, in our research the patients had the opportunity to make personal contact with the professional staff.

Point 14:

Tables 1 and 2 show that the Baseline scores of the 2 groups were the same. This is highly improbable. I assume that these are pooled results. Please separate the baseline scores and state them for each of the subgroups, rather than pooling the whole series.

Answers to the questions:

Indeed, there are pooled results, and we can not separate them due to the lack of baseline data of controls. Due to the nature of the procedure, measuring controls’ baseline data separately did not come into consideration, that is why the results were pooled. We have made the necessary changes to the article as requested (Table 1).

Point 15:

Table 3 is duplicating Tables 1 and 2 to a large degree. There might be a better way of demonstrating these statistical differences between the 2 groups.

Answers to the questions:

It indeed seems redundant, however, different aspects are investigated. According to your suggestion new table was prepared from data belongs to Table 1, 2 and 3. New Table 1 presents Baseline, Control group and Intervention group in the same table so data are more comparable in this way.

Point 16:

All the patients might have accessed other “online” information from other sources and toher countries. This is fairly routine behaviour and many of our own patients come to the clinic with printouts of information they have downloaded from all kinds of websites; including endorsed information booklets.. The authors do not appear to have factored this possibility into their analyses when comparing their 2 cohorts.

Answers to the questions:

We agree with your point, that patients might have accessed other “online” information from other sources. It could be one of the limitation of our research, but our patients were belonged to the older ages, and in our country the older people use the internet less, that is the reason, why we did not measure this effect. On the other hand, we think that, both cohorts had the same opportunity to find other sources.

Reviewer 2 Report

The research aim is to develop a new printed patient educational material in order to reduce fear of surgery before arthroplasties and analyzes its utility in practice.

The abstract is structured appropriately.  

The introduction transposes the research into the topic and formulates the objective of the study at the end. However,  more information about the epidemiology and  the evolution trend of THR and TKR  should be added to highlight their importance, in relation to other scientific papers, for e.g. Moldovan F, Moldovan L, Bataga T. A Comprehensive Research on the Prevalence and Evolution Trend of Orthopedic Surgeries in Romania. Healthcare (Basel). 2023 Jun 27;11(13):1866. doi: 10.3390/healthcare11131866.

In the methodology section, the stages of the research are presented, and the results are clearly described. Wilcoxon rank-sum is the same thing as Mann–Whitney U test, it is not necessary to state when describing the results  (see tables description for e.g.)

The discussions interpret the research results and relate them to other results from the scientific literature.

The conclusions are concise and clear.

The references are adequate, but need proper editing and can be extended as suggested above.

There are some minor editing errors:

·         In the heading of the paper please decide Type of paper (Article, Review, Communication, etc.)

·         The affiliations of the authors should be stated

·         The references should be noted with “[ ]” rather than “( )”

·         The tables in the paper should be provided in APA style

·         The subsections from Methodology and Results should be in italics and numbered for e.g. “2.1 Target Group

Author Response

Response to Reviewer 2 Comments

Dear Reviewer! Thank you for your careful review, for your important comments and suggestions and for giving us the opportunity to resubmit a revised version of the manuscript “Efficiency of printed Patient Information Leaflet written for total knee and hip arthroplasty patients to reduce their fear of surgery” for publication in the Special Issue of Geriatrics.

Point 1:

In the heading of the paper please decide Type of paper (Article, Review, Communication, etc.)

  • Response : We have made the necessary changes in the heading of the paper as requested.

Point 2:

 The affiliations of the authors should be stated.

  • Response : We have made the necessary changes to the article as requested.

Point 3:

 The references should be noted with “[ ]” rather than “( )”

  • Response : We have made the necessary changes to the article as requested.

Point 4:

The tables in the paper should be provided in APA style

  • Response : We have made the necessary changes to the article as requested.

Point 5:

The subsections from Methodology and Results should be in italics and numbered for e.g. “2.1 Target Group

  • Response : We have made the necessary changes to the article as requested.

Round 2

Reviewer 1 Report

Efficiency of printed Patient Information Leaflet written for total knee and hip arthroplasty patients to reduce their fear of surgery.

The authors aimed to establish whether giving patients printed educational material prior to Total Knee Arthoplasty (TKA) and Total Hip Arthroplasty (THA) surgery, might reduce their fear of surgery; which might then in turn help with their post-operative recovery

The Surgical Fear Questionnaire (SFQ) was used to assess fear reduction in the 44 Study patients who received printed information, and 44 “Controls”.  Patients also completed a “self-designed” assessment questionnaire to measure the effectiveness of the leaflet.

The authors found that the SFQ scores decreased significantly in TKA who had been given the printed material (median 37.50 IQR 30.00-40.00 vs. median 20.00 IQR 16.00-24.00) and similarly in the THA patients (median 34.50 IQR 28.00-42.00 vs. median 20.00 IQR 16.00-22.00) compared with the Control Group (median 37.50 IQR 30.00-40.00 vs. median 64.50 IQR 54.00-82.00) and THA (median 34.50 IQR 28.00-42.00 vs. median 73.00 IQR 56.00-81.00). The patients rated the information booklets as “almost entirely usable”.

The authors concluded that the introduction of their new printed patient education material appeared to reduce the fear of surgery.

I reviewed the paper at its first submission, and the authors have adapted their manuscript accordingly. The paper reads better, however some of my queries have not been addressed.

 How were the patients randomised to each group?  (Methodology Section) Can the authors state the study time period, dates etc, how long before surgery these printed material were given, and how long after surgery the patients completed their questionnaires. Did the printed information contain diagrams? How would the authors tackle patients with learning difficulties?

All the patients might have accessed other “online” information from other sources and other countries. This is fairly routine behaviour and many of our own patients come to the clinic with printouts of information they have downloaded from all kinds of websites; including endorsed information booklets.. The authors do not appear to have factored this possibility into their analyses when comparing their 2 cohorts.

Minor language corrections

Author Response

Response to Reviewer 1 Comments

Dear Reviewer! Thank you for your careful review, for your important comments and suggestions and for giving us the opportunity to resubmit a revised version of the manuscript “Efficiency of printed Patient Information Leaflet written for total knee and hip arthroplasty patients to reduce their fear of surgery” for publication in the Special Issue of Geriatrics.

Point 1:

How were the patients randomised to each group?  (Methodology Section)

Response:

  • We have made the necessary changes to the article as requested.

Point 2:

Can the authors state the study time period, dates etc, how long before surgery these printed material were given, and how long after surgery the patients completed their questionnaires?

Response:

  • We have made the necessary changes to the article as requested.

Point 3:

Did the printed information contain diagrams?

Response:

  • We have made the necessary changes to the article as requested.

Point 4:

How would the authors tackle patients with learning difficulties?

Response:

  • We have made the necessary changes to the article as requested.

Point 5:

All the patients might have accessed other “online” information from other sources and other countries. This is fairly routine behaviour and many of our own patients come to the clinic with printouts of information they have downloaded from all kinds of websites; including endorsed information booklets.. The authors do not appear to have factored this possibility into their analyses when comparing their 2 cohorts.

Response:

  • We have made the necessary changes to the article as requested.

Point 6:

Comments on the Quality of English Language: Some minor editing required

Response :

  • The manuscript has been reviewed by an official language proofreader in Hungary.
